# Health Risk Assessment of Heavy Metals and Lipid Quality Indexes in Freshwater Fish from Lakes of Warmia and Mazury Region, Poland

**DOI:** 10.3390/ijerph16193780

**Published:** 2019-10-08

**Authors:** Joanna Łuczyńska, Beata Paszczyk

**Affiliations:** Faculty of Food Sciences, University of Warmia and Mazury in Olsztyn, ul. Plac Cieszyński 1, 10726 Olsztyn, Poland; paszczyk@uwm.edu.pl

**Keywords:** organs of fish, metals, target hazard quotient, dietary intake, lipid quality indexes

## Abstract

The objectives of study were to determine heavy metals content (Zn, Cu, Mn, Fe and Hg) and fatty acids in selected organs of roach, *Rutilus rutilus* (L.); bream, *Abramis brama* (L.); pike, *Esox lucius* (L.); Eurasian perch, *Perca fluviatilis* (L.) collected from reservoirs of Warmia and Mazury region (northeastern Poland). Heavy metals were determined with atomic absorption spectrometry AAS. The fatty acids were analyzed using gas chromatography. In a few cases, differences in the content of heavy metals and fatty acids were not significant between species. The muscles of fish characterized significantly higher values of mercury than other organs (*p* ≤ 0.05), except for bream. The reverse regularity was observed in the case of content of Cu, Zn, Mn and Fe. Fatty acids having a desirable dietary effect in humans (DFA-Hypocholesterolaemic fatty acids) (74.00–74.84) were more than OFA (hypercholesterolaemic fatty acids), i.e., those undesirable (24.03–24.79). The lipid quality indexes AI (index of atherogenicity) (0.40–0.44) and TI (index of thrombogenicity) (0.18–0.24) in muscles of fish were low, which means that the meat of the fish may be recommended for human health. THQ (target hazard quotient) and HI (hazard index) as individual foodstuff were below 1, whereas HI for a specific receptor/pathway combination exceeded 1. This may suggest that eating meat from a given species is safe from a health point of view.

## 1. Introduction

A properly balanced diet, including the amount of fish consumed, is a way to maintain the health and balance of the human body, as well as physiological functions [1]. Fish and fish products play a major role from a nutritional point of view because they are a rich source of nutrients, and provide a relatively low caloric content [2]. Moreover, fish flesh has a good digestibility [3]. According to Pal et al. [2], fish also provide a good balance of protein, and their availability and affordability is better than other animal protein sources [4]. It is also known that fish and fish oils (along with some vegetable oils) are a good source for essential polyunsaturated fatty acids in human nutrition [3,5]. Many clinical trials have shown the relationship between the intake of n-3 PUFA (polyunsaturated fatty acids) and beneficial effects in different diseases (optimal cardiovascular system, brain and vision functioning, cancer prevention, arthritis, hypertension and diabetes mellitus) [2,5,6,7,8]. Since both linoleic acid (LA) and α-linolenic acid (ALA) cannot be synthesized in the body, they are therefore essential [9,10,11]. LA can be converted via γ-linolenic acid (18:3n-6) and dihomo-γ-linolenic acid (20:3n-6; DGLA) to arachidonic acid (20:4n-6; AA). ALA is the precursor for synthesis of eicosapentaenoic acid (20:5n-3; EPA) and further to docosapentaenoic acid (22:5n-3; DPAn-3) and docosahexaenoic acid (22:6n-3; DHA) [12]. However, among fatty acids, there are some saturated fatty acids (SFA) that have different effects on the content of plasma lipoprotein cholesterol fractions. These include to lauric (C12:0), myristic (C14:0) and palmitic acids (C16:0), which increase LDL cholesterol, whereas stearic has no effect. Therefore, replacing SFA (C12:0–C16:0) with polyunsaturated fatty acids (PUFA) decreases LDL cholesterol concentration and the ratio of total/HDL cholesterol. A similar, but lesser effect, is achieved by replacing these SFA with monounsaturated fatty acids (MUFA) [13]. It is clear that saturated fatty acids are hypercholesterolemic fatty acids and monounsaturated fatty acids are neutral or mildly hypocholesterolemic, whereas polyunsaturated fatty acids show the most potent hypocholesterolemic effects [14].

The numerous health benefits provided by fish consumption may be compromised by the presence of undesired substances [15]. The fish take up metals not only from the water, but also from the diet. Thus, the rate of metal accumulation by various aquatic biota and the type and length of food chains determine the amount of dietary metal absorbed by fish [16]. In addition, the amount of metals in the fish body is also influenced by such factors as the type of fish, season, reproductive stage and age, which affect the amount of nutrients and contaminants within a single species as well as between species [15]. These heavy metals pose a danger as chemical water pollution due to their bioaccumulation potential and biomagnification, but it is also known that they cannot be eliminated from the body by metabolic activities [17]. The studies conducted by these authors [17] also showed that heavy metals cause severe damage to fish, thus endangering fish health and the ecosystem. They pose a threat to human health as well via the consumption of heavy metal contaminated fish. Therefore, contaminated meat has no economic value because it is too toxic for consumers [1]. These metals belong to the group of “trace elements”, and their concentration in the body ranges from 0.00001% to 0.01% (e.g., Fe, Zn, Cu and Mn) and they are involved in natural quantities in the regulation of vital functions at all stages of development of the living organism. Iron participates in a wide variety of metabolic processes, including oxygen transport, DNA synthesis and electron transport [18]. Zinc is a component of numerous enzymes, hormone life insulin, growth hormone and sex hormone. This metal is the second trace mineral in the body after iron [19]. Manganese acts as a cofactor for a variety of enzymes, including arginase, glutamine synthetase, pyruvate carboxylase and Mn superoxide dismutase. Through these metalloproteins, this element plays critically important roles, among others, in antioxidant defense, development, digestion and reproduction [20]. Copper-like iron and manganese is a cofactor for numerous enzymes. In addition, it plays an important role in central nervous system development and is required for cellular respiration, neurotransmitter biosynthesis, peptide amidation, pigment formation and connective tissue strength [21]. The other group is “ultratrace elements”, and their concentration is lower than 0.000001% (e.g., Hg) [22]. Mercury may cause both toxicological cellular and cardiovascular, hematological, pulmonary, renal, immunological, neurological, endocrine, reproductive and embryonic toxicological effects [23].

Therefore, the aim of this study was:

1) The identification of the differences between mercury, zinc, iron, copper and manganese concentration in four freshwater fish species, and their organs (muscles, liver and gills), as well as the impact of biometric parameters (body weight and total length) in fish organs.

2) The identification of the differences between fatty acids and the lipid quality indexes in muscles of examined fish.

3) To examine the health risk assessment using estimated daily intake (EDI), hazardous index (HI), target hazard quotient (THQ) and other indexes.

## 2. Materials and Methods

### 2.1. Sampling and Sample Preparation

Predatory (European perch, *Perca fluviatilis* L. and pike, *Esox lucius* L.) and non-predatory fish (roach, *Rutilus rutilus* L and bream, *Abramis brama* L.) were caught from the Olsztyn lake District (October 2013) (Figure 1). The fish were weighed and measured after being brought to the laboratory. Muscles (without skin) from the dorsal part, liver and gills were prepared from one specimen and stored until analysis in the refrigerator at −30 °C. To avoid sample contamination, all organs were collected with a plastic knife and stored in sealed plastic bags.

Ethical permit: fish were bought at the fish farm and were already dead. According to European and Polish Law, the research done on the tissue of commercially caught fish is free from obtaining permission from the Local Ethical Commission.

### 2.2. Element Analysis

#### 2.2.1. Mercury

Duplicate samples of organs were weighed into quartz boats (<270 mg ± 0.0001 g). The total mercury was processed with atomic absorption spectrometry thermal decomposition using Milestone DMA-80 with dual-cell and UV enhanced photodiodes (Italy). Conditions for the determination of mercury is described in an earlier publication [24].

#### 2.2.2. Zinc, Copper, Iron and Manganese

For the determination of zinc, copper, iron and manganese in muscles of fish examined, samples were ashed at 450 °C using laboratory furnaces (Nabertherm, Germany). After obtaining the white ash, it was dissolved in 1 M HNO_3_ (Suprapur-Merck, Darmstadt, Germany) and transferred with deionized water (Merck-Millipore Elix Advantage 3, USA) into a volumetric flask of volume 25 mL. After weighing, samples of liver and gills were placed in a boro-silica glass tube and mineralized with a nitric and perchloric acid mixture (Merck, Darmstadt, Germany, 3:1, v/v) at 190 °C. For this purpose, a heating block with a programmable temperature and digestion time was used (DK 20, VELP Scientifica, Italy). The resulting solution was transferred to flasks with a volume of 25 mL, using deionized water.

### 2.3. Instrumental Analysis and Quality Control

Samples were prepared in two parallel wells. The total mercury was determined using Milestone DMA-80 with dual-cell and UV enhanced photodiodes (Italy). The detection limit (LOD) was 0.02 μg/kg. The quality control of methods was tested using the reference material: BCR CRM 422 (muscles of cod, *Gadus morhua* (L.)) with a certified value of mercury. The recovery rate of Hg was 100.2% (*n* = 4).

Concentrations of zinc, copper, iron and manganese were measured using the flame atomic absorption spectrometry (iCE 3000 Series AAS, Thermo Scientific, Loughborough, England). Four blanks and four standards were analyzed with each batch of samples. The absorptions wavelength was as follows: 213.9 nm for zinc and 324.8 nm for copper. The calibration curves were prepared using four solution standards (1000 μg/L) with 0.1 M HNO₃ supplied by J.T.Baker^®^ (Deventer, Netherlands). The calibration curves were linear within the range of metal concentration (regression coefficients R^2^ ≥ 0.999). The absorption wavelengths were as follows: 248.3 nm for iron, 213.9 nm for zinc, 324.8 nm for copper, 279.5 nm for manganese. The detection limits (LOD) were 0.5 mg/kg for Fe, 0.1 mg/kg for Zn, 0.05 mg/kg for Cu and 0.05 mg/kg for Mn. Sensitivity was as follows: 0.05, 0.05, 0.02 and 0.02 mg/L. The quality control of methods was tested using the elements in reference material: BCR CRM 422 (muscles of cod *Gadus morhua* (L.)) with a certified value of zinc, copper and mercury. The recovery rates were: 105.0% Zn, 103.0% Cu. 96% Fe and 103% Mn, respectively.

### 2.4. Fatty Acids Analysis

The lipids were extracted according to the Folch’s procedure [25]. The fatty acid methyl esters were prepared from total lipids with the Peisker method with chloroform, methanol and sulphuric acid (100:100:1 v/v) [26].

The fatty acids of methyl esters of each sample were analyzed using 7890A Agilent Technologies chromatograph with a flame-ionization detector (FID) (Agilent Technologies, INC., Santa Clara, California, USA) under the following conditions: capillary column (dimension 30 m × 0.25 μm with a 0.32 mm internal diameter, liquid phase StabilwaxR), temperature: flame-ionization detector −250 °C, injector −230 °C, column −190 °C, carrier gas—helium with a flow rate 1.5 mL/min. Individual fatty acids were identified by comparing the relative retention time peaks to the known Supelco’s standards.

### 2.5. The Lipid Quality Indexes (AI and TI)

They were calculated using the following pattern by Ulbricht and Southgate [27] and Garaffo et al. [28]:

Index of atherogenicity (AI):AI = [C12:0 + (4 × C14:0) + C16:0]/(n-3PUFA + n-6PUFA + MUFA)] (1)

Index of thrombogenicity (TI):TI = [C14:0 + C16:0 + C18:0]/[(0.5 × C18:1) + (0.5 × sum of other MUFA) + (0.5 × n-6PUFA) + (3 × n-3PUFA) + n-3PUFA/n-6PUFA)] (2)

Flesh-lipid quality (FLQ) were calculated presented by Abrami et al. [29] and Senso et al. [30]:FLQ = 100 × (EPA + DHA)/(% of total fatty acids) (3)

Hypercholesterolaemic fatty acids (OFA):OFA = C12:0 + C14:0 + C16:0 (4)

Hypocholesterolaemic fatty acids (DFA):DFA = C18:0 + UFA (5)

### 2.6. Human Health Risk Assessment

#### 2.6.1. Estimated Daily Intake of Mercury (EDI)

EDI is the estimated daily intake (μg/kg body weight/day), and is obtained as follows:EDI = C × IR/BW (6)
where C is the average concentration of mercury in food stuffs (μg/g wet weight), IR is the daily ingestion rate (g/daily) and BW is the average body weight (60 kg) [31].

#### 2.6.2. Target Hazard Quotient (THQ)

According to Ahmed et al. [32] and US EPA [33], THQ estimated the non-carcinogenic health risk of consumers due to intake of heavy metal contaminated fish use an oral reference dose of Hg, Zn, Cu, Fe and Mn (RfD = 3.00 × 10^−4^, 3.00 × 10^−1^, 7.00 × 10^−1^, 1.4 × 10^−1^). When THQ < 1, there is health benefit from fish consumption and the consumers are safe, whereas THQ > 1 suggests a high probability of adverse risk of human health.
THQ = (EFr × ED × FiR × C/RfD × BW × TA) × 10^−3^(7)
where Efr is the exposure frequency (365 days/year), ED is the exposure duration (70 years), FiR is the fish ingestion rate (g/person/day), C is the average concentration of mercury in food stuffs (μg/g wet weight), RfD is the oral reference dose (mg/kg/day) (US EPA 2017), BW is the average body weight of local residents (60 kg) [31], TA is the average exposure time (365 days/year × ED).

#### 2.6.3. The Combined Risk of Many Heavy Metals

The TTHQ of heavy metals for individual foodstuff was treated as the mathematical sum of each individual metal THQ value [34]:TTHQ individual foodstuff = THQ (toxicant 1) + THQ (toxicant 2) + ……+ THQ (toxicant n) = THQ(Hg) + THQ(Cu) + THQ(Zn) + THQ(Fe) + THQ(Mn)(8)
HI for a specific receptor/pathway combination (e.g., diet) was calculated using the following pattern [34]:HI = TTHQ (foodstuff 1) + TTHQ (foodstuff 2) + TTHQ (foodstuff 3) + TTHQ (foodstuff 4)(9)
When the HI exceeds 1, there may be concern for potential health risks.

### 2.7. Statistical Analysis

Significant differences in the content of fatty acids and lipid quality indexes in the muscle tissue of fish studied were estimated using a one-way analysis of variance (ANOVA). Similar, differences in the concentration of heavy metals, between species and organs of the same species were calculated using STATISTICA 13.1 program (StatSoft, Kraków, Poland). Differences were found to be significant at *p* < 0.05. When the Bartlett’s test showed that the variances were heterogeneous, values in particular groups were transformed (log x). The correlation coefficients between the content of heavy metals in organs of fish and their size (body weight and total length) were calculated using a STATISTICA 13.1 software. The significance level of *p* < 0.05 was used.

## 3. Results and Discussion

### 3.1. Differences in the Content of Heavy Metals

The muscles of fish examined had significantly higher values of mercury than other organs (*p* ≤ 0.05), except for bream, which had no statistically significant differences between the content of mercury in the muscles and liver of this species (*p* > 0.05) (Table 1). Reverse regularity was observed for the content of Cu, Zn, Mn and Fe. The liver of the examined fish contained more Cu than other organs (*p* ≤ 0.05). The liver of roach and perch had a significantly higher content of Fe, whereas bream liver had more Zn (*p* ≤ 0.05). There were no significant differences between the content of Fe in the liver and gills of two species (bream and pike) and between the concentration of Zn in the liver and gills of perch. In other cases, higher contents of marked metals were observed in the gills of the examined fish (*p* ≤ 0.05).

In most cases, it was not found that the feeding type of fish had a significant effect on the content of such elements as Cu, Zn, Fe and Mn (Figure 2). Predatory fish (perch and pike) which occupy the last link in the trophic chain of the aquatic ecosystem contained more mercury in the muscles and liver than those of the studied fish belonging to the lower trophic levels (bream and roach) (*p* ≤ 0.05). This regularity was not found in gills of the examined fish (*p* > 0.05).

Generally, the concentration of 18 elements, including Hg, Fe, Cu, Mn and Zn significantly differed between species such as sichel (*Pelecus cultratus*), ruffe (*Gymnocephalus cernua*) and European perch (*Perca fluviatilis*), caught at a polluted segment of the Danube River near Belgrade (*p* < 0.0001) [35]. Similarly, Sayegh Petkovšek et al. [36] found that fish (*Abramis brama* danubii, *Alburnus alburnus alburnus*, *Barbus meridionalis petenyi*, *Carassius auratius* gibelio, *Cyprinus carpio*, *Lepomis gibossus*, *Leuciscius cephalus* cephalus, *Perca fluviatilis*, *Rutilus rutilus* and *Scardinus erythrophtlalmus erythrophtlalmus*) collected in the Šalek lakes (Slovenia) could be used to assess the bioavailability of metals in freshwater biota, although the concentration of metals (e.g., Hg and Zn) varied significantly between fish species. To compare the current results with the research of other authors, Table 2 presents the content of heavy metals of selected fish organs from various aquatic ecosystems located in Europe [35,36,37,38,39,40,41,42,43,44]. Based on these data, it was found that in most cases, these results did not coincide with the results covered by the study of this work. First of all, the mercury content does not accumulate along the trophic chain, except for Hg in the muscles of fish caught from the Kirchera River [37]. In addition, the fish muscles (Table 2) did not contain more mercury than the other examined organs presented in the table. The authors also did not find clear relationships between the content of Cu, Mn, Zn and Fe and the length of the trophic chain of a given water reservoir. Some relationships, where the amount of Zn, Cu and Mn was higher in fish organs inhabiting lower levels of the chain, were observed in fish from the West Morava River Basin [38].

### 3.2. Correlation between Metal Pairs and Size

Figure 3 only shows the significant correlation between metals in fish organs. In other cases, no significant correlations were observed. Significant positive correlation coefficients were found between metal pairs in muscles of perch (Fe-Mn, Hg-Cu), pike (Hg-Zn) and roach (Cu-Mn, Fe-Zn, Fe-Mn), gills of perch (Cu-Zn, Cu-Mn, Zn-Mn), pike (Cu-Mn), roach (Cu-Zn, Fe-Cu, Fe-Mn) and bream (Hg-Cu) and liver of roach (Fe-Zn).

Significant positive correlations were found for Zn-Fe, Zn-Mn, while significant negative correlations were observed for Zn-Cu in the muscles of fish (*Scardinus knezevici, Alburnus scoranza, Cyprinus carpio, Rutilus prespensis, Anguilla anguilla and Perca fluviatilis*) collected from Skadar Lake (Montenegro) [39]. The results obtained by these authors are consistent with the results of the current study, but only for roach muscles. However, the current results are in contradiction with previous studies received by Łuczyńska and Tońska [45].

Figure 4 and Figure 5 only shows the significant correlation between content of metals and size of fish. In other cases, no significant correlations were observed. The content of Hg in muscles of all fish species was positively correlated with body weight and total length (*p* < 0.018) (Figure 4 and Figure 5). Only the concentration of Hg in gills and liver of perch increased along with increased body weight (*p* = 0.000 and *p* = 0.004) (Figure 4) and length (*p* = 0.008 and *p* = 0.002), respectively (Figure 5). Significant positive correlation were also found between the content of Cu in the muscle tissue of perch and body weight (*p* = 0.001) and length (*p* = 0.010), whereas the content of Cu in gills of the same species decreased as body weight (*p* = 0.05) and total length (*p* = 0.003) increased. Significant positive correlations between body weight (*p* = 0.014) (Figure 4) and total length (*p* = 0.020) (Figure 5) and the content of Zn were found in muscles of pike. There is a negative correlation between the content of Zn, Mn and Fe in muscles of roach and total length (*p* = 0.037, *p* = 0.031, *p* = 0.011, respectively) and between the level of Fe and their body weight (*p* = 0.043) (Figure 4). Negative correlation factors between Cu content in gills of roach and body weight (*p* = 0.048) and length (*p* = 0.019) were noted. In the case of bream, the content of zinc in gills decreased as body weight increased (*p* = 0.008) (Figure 4). Similarly, Zn in gills and Fe in liver of bream decreased as total length increased (*p* = 0.017, *p* = 0.027, respectively). The obtained results could have been influenced by the variety of conditions of the aquatic environment in which the fish lived, related to the water reaction and the availability of metals.

There were no significant correlations between Fe and Hg in muscles of perch (Skandar Lake, Montenegro) and their age or size (length and weight) [39]. However, positive correlation between Hg in muscles of pike, perch, roach, bream and their size were found in previous studies [24,46,47]. Nozari et al. [48] found a positive correlation between these parameters in muscles of pike from Anzali International Wetland (Iran), while Miller et al. [49] reported a similar correlation in muscles of perch from Swedish and Finnish aquatic environments. This is accordance with the current results (Figure 4 and Figure 5). According to Łuczyńska et al. [47] zinc in the gills of roach was negatively correlated with their weight (r = −0.693, *p* = 0.026) and length (r = −0.668, *p* = 0.035). During the study, the correlation was also negative, but not statistically significant (Figure 4 and Figure 5). The same authors found a positive correlation between the Zn content in gills of perch and their weight (r = 0.634, *p* = 0.049) and no statistically significant correlation between the copper content in the roach and perch organs and their size [47]. This is in contradiction with the results being studied in the current study (Figure 4 and Figure 5). 

### 3.3. Fatty Acids and Lipid Quality Indexes

Among the species under study, pike had a significantly higher n-3/n-6 ratio (3.56) and Σ n-3 PUFA (42.61%) than the other fish species (*p* ≤ 0.05) (Table 3). There were no significant differences between the values of Σ SFA (31.73–32.94%) and Σ n-6 PUFA (12.2–14.98%) in muscles of the examined fish. Σ MUFA (24.12%) in muscles of bream was significantly higher (*p* ≤ 0.05) than other fish species, with the exception of roach. Among the SFA and MUFA, the predominant fatty acids were palmitic acid C16:0 and oleic acid C18:1, whereas arachidonic C20:4 n-6, eicosapentaenoic EPA n-3 and docosahexaenoic DHA n-3 acids were the most abundant of PUFA.

Atherogenic index (AI), hypocholesterolemic (OFA) and hypercholesterolemic fatty acids (DFA) in muscles of the examined fish did not differ significantly (*p* > 0.05) (Figure 6). The flesh-lipid quality index (FLQ) in the muscles of fish included in this study gave rise to the following sequence: pike > perch ≈ roach > bream (*p* ≤ 0.05). No difference was found in terms of thrombogenicity index (TI) (Figure 6) between the species within the following groups (bream, roach and perch) and (roach, pike and perch) (*p* > 0.05). The low values of AI and TI indicates that the tissue of the studied fish is beneficial from a health point of view. Fatty acids with undesirable dietary effect in humans (OFA) were less than DFA, i.e., those desirable. As with heavy metals, there was no clear relationship between the tested fatty acids or selected quality indicators and the fish species or their occurrence in organs and biometric data.

Many studies indicate that the fatty acids that dominated the muscles of the fish in the current study are the same groups of fatty acids found in the muscles of the species of fish in other studies [50,51,52,53,54,55]. Kainz et al. [56] found that n-3 and n-6 PUFA in fish of pre-alpine Lake Lunz (Austria) decreased with increasing trophic position, demonstrating that these essential fatty acids did not biomagnify with increasing trophic level. These observations are not consistent with the results of the current study (Table 3). Previous studies conducted by Łuczyńska et al. [51] also found that n-3 and n-6 PUFA did not differ between predatory and non-predatory fish. Based on the results, it can be stated that the values of these indices were not affected by the level of the trophic chain and, thus, the diet of the fish. Łuczyńska et al. [57], did not find any significant differences either between these indices for fish such as bream and perch, which occupy two different links in the aquatic ecosystem food chain. The results of the current research on AI and TI (Figure 6) are consistent with studies conducted by Linhartová et al. [58] which found that the indicators in all analyzed fish, except for Nile tilapia, were below 0.5, which indicates significant benefit to human health if these fish are included in the human diet. AI and TI for seven freshwater fish species, including perch and bream, from the Czech Republic were close to the values given for the Eskimo diet, which indicates the high nutritional value of these fish and, therefore, not a threat [59]. According to Linhartová et al. [58] the higher values of these coefficients, the higher risk of developing cardiovascular diseases, because AI indicates the risk of diseases such as atherosclerosis (deposition of fat in the walls of the arteries) and TI determines the possibility of blood clots. Tilami et al. [59] showed that AI and TI for bream and perch were 0.30 and 0.38 or 0.25 and 0.22, respectively. Similar values were found in these species of fish subject to this study (Figure 6).

### 3.4. Human Health Risk

Fish consumption in 2013 year (Poland) was 12.2 kg per capita/year (for an adult with a body weight of 60 kg) [60]. Therefore, the estimated daily intake (EDI) of Cu, Zn, Mn, Fe and Hg from the 33.42 g/person/day were: 0.076–0.117, 2.330–5.582, 0.027–0.068, 0.489–0.729 and 0.036–0.193, respectively (Table 4). The THQ and HI as individual foodstuff (TTHQ) were below 1 (Table 5), whereas HI for a specific receptor/pathway combination (e.g., diet) exceeded 1. This may suggest that eating meat from a given species is a non-carcinogenic health risk for consumers.

Łuczyńska et al. [57], studied European perch and bream from the Polish market, and found that the THQ values were also below 1 and there was no carcinogenic health risk to the consumer by consuming fish. According to these authors the daily intake (EDI) of mercury was: 0.009 μg/body weight (bream) and 0.076 (perch). EDI of mercury from the portion of fish examined was higher (Table 4). THQ and HI of Cu, Zn and Hg in European perch and roach from Lake Pluszne were lower than 1, which is important for the local community and other people who use the lake for recreational purposes [42], although the EDI of zinc and copper from the 33.698 g portions of fish was higher and the EDI of mercury was lower than the examined fish (Table 4). According to Addo-Bediako et al. [61], when THQ < 1, these adverse health effects are unlikely, whereas THQ > 1 suggests a high probability of adverse health effects. A THQ value (for Mn and Zn) below 1 in the roach muscles from the Miankaleh international wetland was also obtained by the Alipour et al. [42]. There was also no threat from metals, including Cu, Fe, Mn and Zn, in the consumption of fish living in lakes, rivers, streams, and the sea in Sakarya (Turkey) [62]. This is accordance with the results obtained in the current study. On the basis of the 18 elements under study, the authors observed that both bream and zander are among the best fish species to monitor the quality of fish tissue, but fisheries monitoring programs should also include other economically important fish species (e.g., common carp, pike and barbell) and some exotic species, such as Prussian carp [63].

## 4. Conclusions

Based on the hazard factor for individual metals THQ < 1, it was found that the consumption of the studied fish does not constitute a carcinogenic health risk. Similarly, when the mathematical sum of each THQ value for a given species of fish is considered, their consumption does not raise concerns related to adverse health risks. However, when a specific receptor/pathway combination (e.g., diet) was calculated, which posed a threat from the consumption of all fish species simultaneously contaminated with test elements, there could be concerns about potential health risk because HI was below 1. Looking at the low daily intake of fish, it appears that these fish are safe from a nutritional and health point of view. It was also observed that these fish are a rich source of fatty acids and the proportions between fatty acids with desirable dietary effects (DFA) and those with undesirable effects (OFA) were in favor of the former. In addition, AI and TI lipid quality indices in fish muscles were low, which means that fish tissue is beneficial to consumers. There was no clear relationship between the tested heavy metals, fatty acids or selected quality indicators and the fish species or their occurrence in organs and biometric data. Such a relationship could be observed only in relation to mercury, which accumulated in fish occupying individual levels of the trophic chain, with humans at the end. However, its levels did not exceed acceptable standards.

## Figures and Tables

**Figure 1 ijerph-16-03780-f001:**
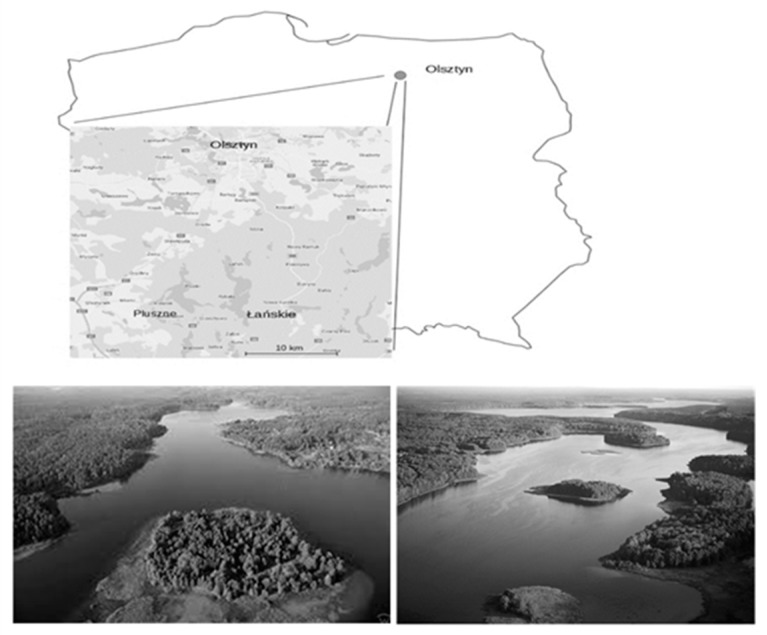
The study area was located in north-eastern Poland, near the city Olsztyn.

**Figure 2 ijerph-16-03780-f002:**
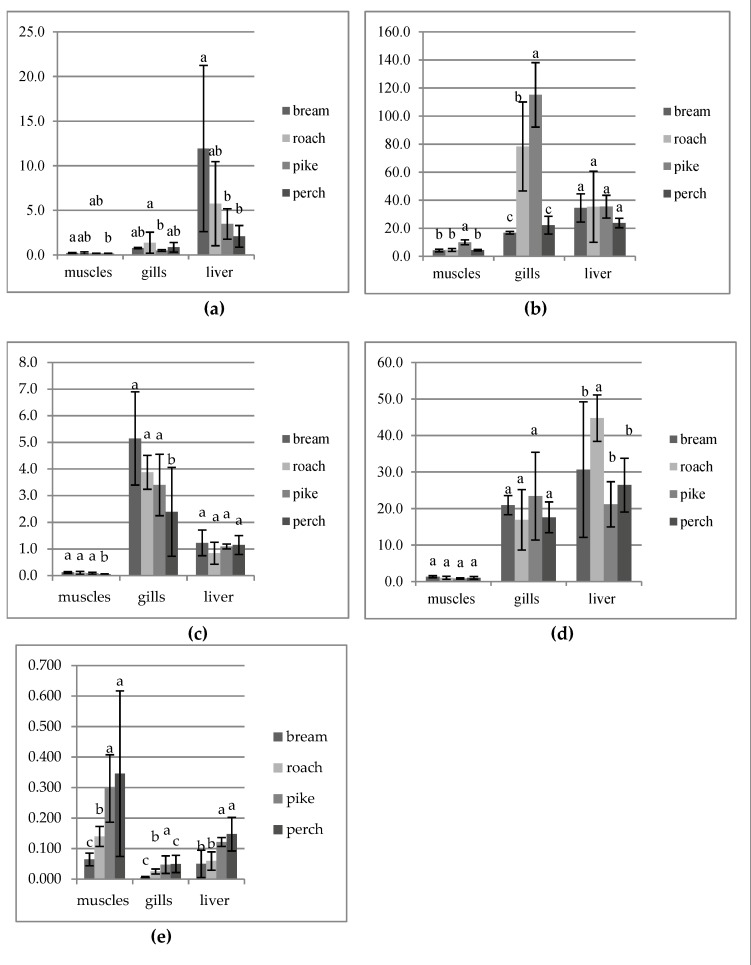
Differences (mean ± SD) in the content of heavy metals in the same organs of fish (**a**) Cu, (**b**) Zn, (**c**) Mn, (**d**) Fe and (**e**) Hg; a, b, c—significant differences between the same organs of the different species (*p* ≤ 0.05). The same letter indicates the absence of significant differences (*p* > 0.05).

**Figure 3 ijerph-16-03780-f003:**
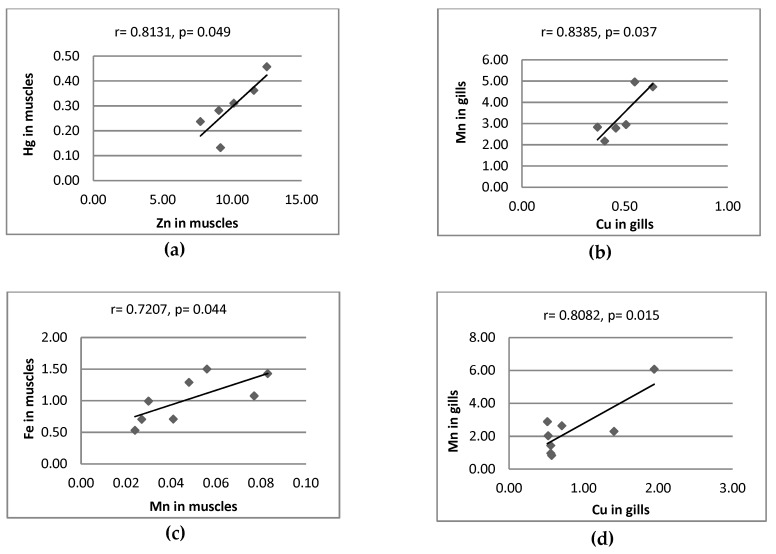
Correlation between contents of metal (mg/kg wet weight); r—correlation coefficients; *p*—significant level; (**a**) pike, (**b**) pike, (**c**) perch, (**d**) perch, (**e**) perch, (**f**) perch, (**g**) roach, (**h**) perch, (**i**) roach, (**j**) roach, (**k**) roach, (**l**) roach, (**m**) roach, (**n**) bream, (**o**) roach.

**Figure 4 ijerph-16-03780-f004:**
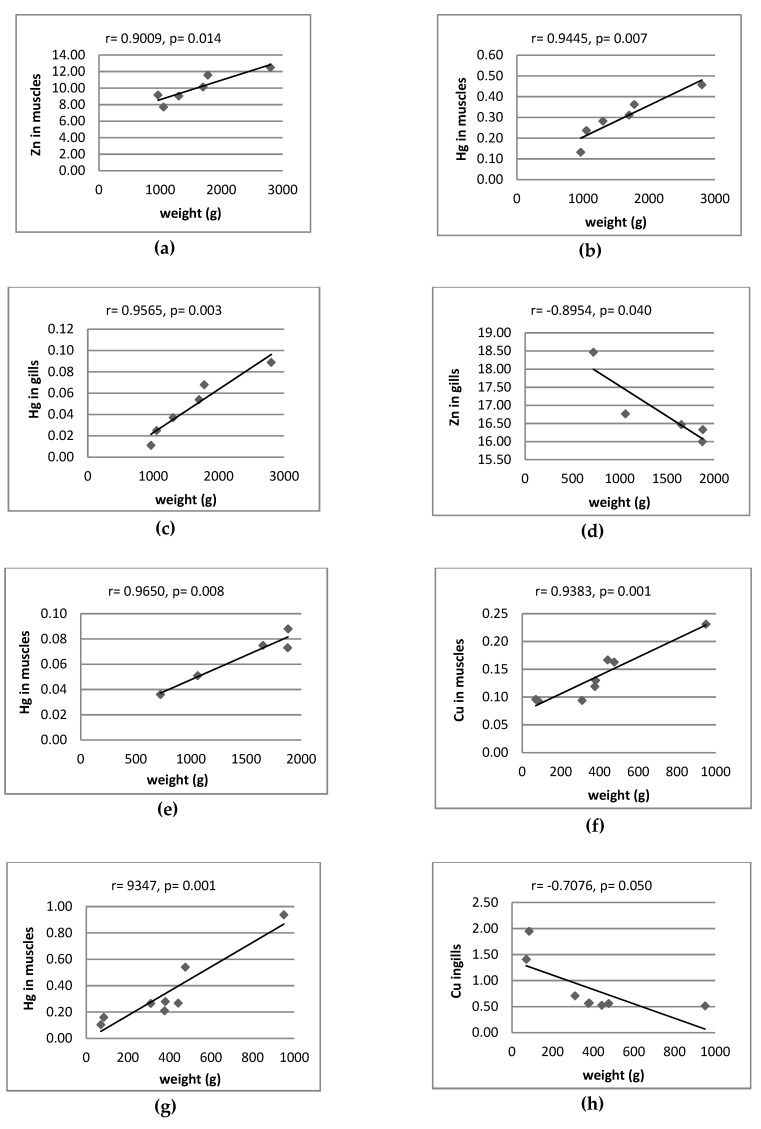
Correlation between contents of metal (mg/kg wet weight) and weight of fish. r—correlation coefficients; *p*—significant level; (**a**) pike, (**b**) pike, (**c**) pike, (**d**) bream, (**e**) bream, (**f**) perch, (**g**) perch, (**h**) perch, (**i**) roach, (**j**) roach.

**Figure 5 ijerph-16-03780-f005:**
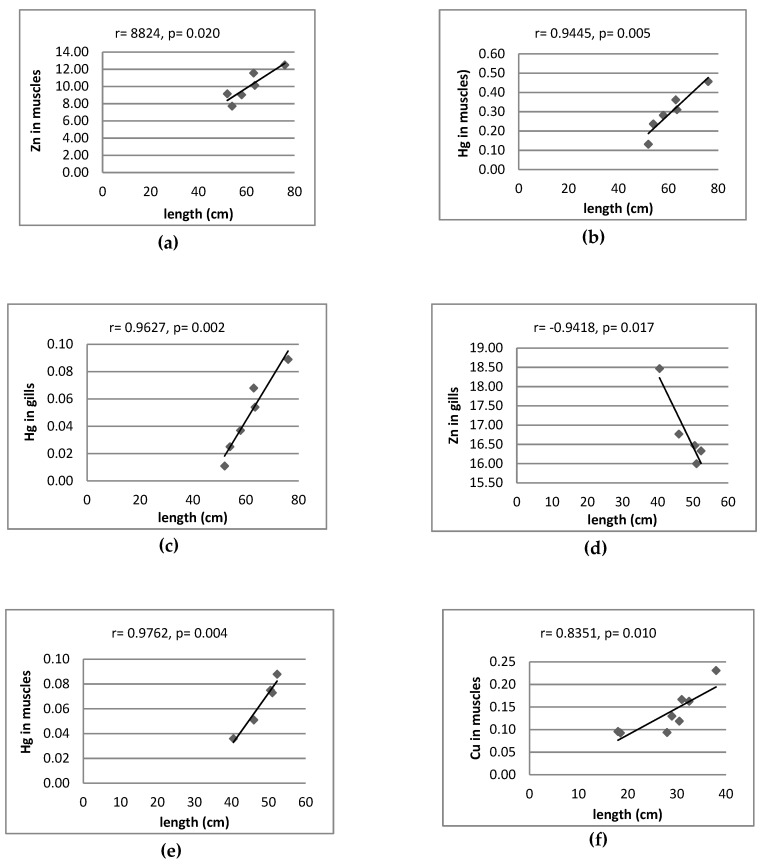
Correlation between contents of metal (mg/kg wet weight) and length of fish. r—correlation coefficients; *p*—significant level; (**a**) pike, (**b**) pike, (**c**) pike, (**d**) bream, (**e**) bream, (**f**) perch, (**g**) perch, (**h**) perch, (**i**) roach, (**j)** roach, (**k**) roach, (**l**) roach, (**m**) roach.

**Figure 6 ijerph-16-03780-f006:**
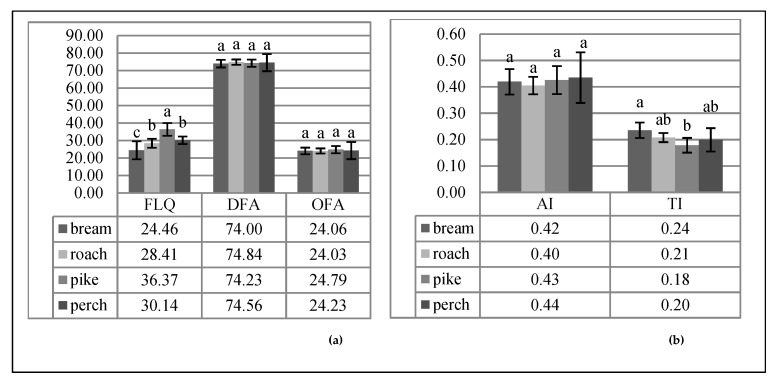
Lipid quality indexes. (**a**) Flesh-lipid quality (FLQ); hypercholesterolaemic fatty acids (OFA); hypocholesterolaemic fatty acids (DFA); (**b**) index of atherogenicity (AI); index of thrombogenicity (TI); a, b, c—significant differences between the fish of the different species (*p* ≤ 0.05). The same letter indicates the absence of significant differences (*p* > 0.05).

**Table 1 ijerph-16-03780-t001:** Interspecific differences (mean ± SD) in the content of heavy metals in the organs of the same fish species.

	Bream *Abramis brama* L. (n = 5)		Roach *Rutilus rutilus* L. (n = 9)		Pike *Esox lucius* L. (n = 6)		Perch *Perca fluviatilis* L. (n = 8)	
	Mean	SD	Mean	SD	Mean	SD	Mean	SD
	Mg/kg Wet Weight							
Length (cm)	48.1	4.8	30.8	7.2	61.1	8.7	28.2	6.8
Weight (g)	1438.0	521.7	426.2	272.9	1601.7	675.3	386.0	274.2
Muscles Cu	0.210 ^c^	0.053	0.208 ^c^	0.127	0.155 ^c^	0.029	0.137 ^c^	0.048
Fills Cu	0.766 ^b^	0.062	1.366 ^b^	1.181	0.486 ^b^	0.099	0.849 ^b^	0.535
Liver Cu	11.92 ^a^	9.313	5.749 ^a^	4.718	3.475 ^a^	1.710	2.085 ^a^	1.211
Muscles Zn	4.183 ^c^	0.915	4.522 ^c^	1.035	10.020 ^c^	1.763	4.352 ^b^	0.520
Gills Zn	16.81 ^b^	0.968	78.33 ^a^	31.67	115.1 ^a^	22.95	22.21 ^a^	6.316
Liver Zn	34.46 ^a^	10.08	35.38 ^b^	25.38	35.41 ^b^	8.095	23.72 ^a^	3.331
Muscles Mn	0.122 ^c^	0.032	0.106 ^c^	0.054	0.090 ^c^	0.039	0.048 ^c^	0.022
Gills Mn	5.148 ^a^	1.751	3.874 ^a^	0.638	3.402 ^a^	1.150	2.394 ^a^	1.666
Liver Mn	1.230 ^b^	0.481	0.843 ^b^	0.413	1.095 ^b^	0.092	1.149 ^b^	0.355
Muscles Fe	1.309 ^b^	0.286	1.005 ^c^	0.435	0.833 ^b^	0.170	1.030 ^c^	0.360
Gills Fe	20.91 ^a^	2.571	16.89 ^b^	8.253	23.38 ^a^	12.01	17.59 ^b^	4.211
Liver Fe	30.65 ^a^	18.56	44.74 ^a^	6.364	21.16 ^a^	6.163	26.40 ^a^	7.352
Muscles Hg	0.065 ^a^	0.021	0.140 ^a^	0.033	0.297 ^a^	0.111	0.346 ^a^	0.271
Gills Hg	0.006 ^b^	0.002	0.025 ^c^	0.008	0.047 ^c^	0.029	0.050 ^c^	0.028
Liver Hg	0.050 ^a^	0.045	0.059 ^b^	0.030	0.121 ^b^	0.015	0.147 ^b^	0.055

n—number of fish; SD—standard deviation; ^a, b, c^—significant difference (*p* ≤ 0.05). The same letter indicates the absence of significant differences between organs of the same fish studied.

**Table 2 ijerph-16-03780-t002:** The content of heavy metals in organs of freshwater fish (mg/kg wet weight).

Species		Hg	Zn	Cu	Fe	Mn	References
**Muscles**							
Roach, *Rutilus rutilus* (L.)	Skalka Reservoir	0.81	-	-	-	-	[40]
Bream, *Abramis brama* (L.)	West Morava River Basin	1.22	27.30	1.45	21.13	0.82	[38]
Roach, *Rutilus rutilus* (L.)	West Morava River Basin	0.95	47.77	1.47	25.51	1.02	[38]
Perch, *Perca fluviatilis* (L.)	West Morava River Basin	2.06	22.69	0.02	9.236	0.45	[38]
Pike, *Esox lucius* (L.)	West Morava River Basin	1.04	48.57	0.84	18.304	1.249	[38]
Bream, *Abramis brama* (L.)	Skalka Reservoir	0.96	-	-	-	-	[40]
Perch, *Perca fluviatilis* (L.)	Velenjsko Lake	0.12	12.5	-	-	-	[41]
Perch, *Perca fluviatilis* (L.)	Šalek lakes	0.12	12.5	-	-	-	[36]
Roach, *Rutilus rutilus* (L.)	Šalek lakes	0.08	13.4	-	-	-	[36]
Bream, *Abramis brama danubii*	Šalek lakes	0.16	7.79	-	-	-	[36]
Roach, *Rutilus rutilus* (L.)	Miankaleh wetland	-	7.2	1.6	28	-	[42]
Perch, *Perca fluviatilis* (L.)	Danube River near Belgrade	2.72	18.89	0.45	11.85	0.69	[35]
Roach, *Rutilus rutilus* (L.)	Kirchera River	0.01	4.81	-	-	0.22	[37]
Perch, *Perca fluviatilis* (L.)	Kirchera River	0.04	4.33	-	-	0.18	[37]
Pike, *Esox lucius* (L.)	Kirchera River	0.07	2.88	-	-	0.11	[37]
Bream, *Abramis brama* (L.)	Żnin Duże Lake	-	1.516	0.889	1.704	0.807	[43]
Roach, *Rutilus rutilus* (L.)	Żnin Duże Lake	-	1.761	0.659	1.519	0.996	[43]
Perch, *Perca fluviatilis* (L.)	Żnin Duże Lake	-	1.546	0.980	1.572	0.768	[43]
Perch, *Perca fluviatilis* (L.)	Skadar Lake	0.121	5.83	0.81	4.77	0.22	[39]
Pike, *Esox lucius* (L.)	Ińsko Lake	0.01	9.4	0.14	1.4	0.24	[44]
Bream, *Abramis brama* (L.)	Miedwie	-	2.3	0.14	1.3	0.11	[44]
Perch, *Perca fluviatilis* (L.)	Miedwie	0.01	4.6	0.14	1.2	0.18	[44]
**Liver**							
Bream, *Abramis brama* (L.)	Skalka Reservoir	1.50	-	-	-	-	[40]
Roach, *Rutilus rutilus* (L.)	Skalka Reservoir	0.88	-	-	-	-	[40]
Bream, *Abramis brama* (L.)	West Morava River Basin	1.05	91.81	44.75	428.11	5.82	[38]
Roach, *Rutilus rutilus* (L.)	West Morava River Basin	0.95	80.51	30.43	177.23	5.05	[38]
Perch, *Perca fluviatilis* (L.)	West Morava River Basin	1.85	71.80	17.41	355.07	2.1	[38]
Pike, *Esox lucius* (L.)	West Morava River Basin	0.81	90.72	13.88	261.38	1.85	[38]
Bream, *Abramis brama* (L.)	Żnin Duże Lake	-	2.098	1.755	2.436	1.143	[43]
Roach, *Rutilus rutilus* (L.)	Żnin Duże Lake	-	2.382	2.169	2.631	1.405	[43]
Perch, *Perca fluviatilis* (L.)	Żnin Duże Lake	-	2.012	1.395	2.358	1.143	[43]
Perch, *Perca fluviatilis* (L.)	Danube River near Belgrade	2.52	77.66	18.20	225.0	4.24	[35]
Perch, *Perca fluviatilis* (L.)	Velenjsko Lake	0.22	29.3	-	-	-	[41]
Perch, *Perca fluviatilis* (L.)	Šalek lakes	0.22	29.3	-	-	-	[36]
Roach, *Rutilus rutilus* (L.)	Šalek lakes	0.09	29.2	-	-	-	[36]
Bream, *Abramis brama danubii*	Šalek lakes	0.31	28.0	-	-	-	[36]
Roach, *Rutilus rutilus* (L.)	Kirchera River	0.02	18.26	-	-	4.09	[37]
Perch, *Perca fluviatilis* (L.)	Kirchera River	0.02	53.56	-	-	2.24	[37]
Pike, *Esox lucius* (L.)	Kirchera River	0.02	36.18	-	-	0.59	[37]
**Gills**							
Bream, *Abramis brama* (L.)	West Morava River Basin	1.31	68.46	2.36	389.61	21.45	[38]
Roach, *Rutilus rutilus* (L.)	West Morava River Basin	1.27	196.69	3.1	180.98	20.57	[38]
Perch, *Perca fluviatilis* (L.)	West Morava River Basin	1.8	71.2	0.99	138.52	12.29	[38]
Pike, *Esox lucius* (L.)	West Morava River Basin	1.27	558.11	1.56	96.69	26.83	[38]
Perch, *Perca fluviatilis* (L.)	Šalek lakes	0.06	24.9	-	-	-	[36]
Roach, *Rutilus rutilus* (L.)	Šalek lakes	0.06	78.7	-	-	-	[36]
Bream, *Abramis brama danubii*	Šalek lakes	0.03	16.5	-	-	-	[36]
Perch, *Perca fluviatilis* (L.)	Danube River near Belgrade	1.84	64.82	0.66	189.39	10.57	[35]
Bream, *Abramis brama* (L.)	Żnin Duże Lake	-	1.937	0.650	2.436	1.745	[43]
Roach, *Rutilus rutilus* (L.)	Żnin Duże Lake	-	3.030	0.956	2.364	1.632	[43]
Perch, *Perca fluviatilis* (L.)	Żnin Duże Lake	-	1.945	0.663	2.212	1.680	[43]
Perch, *Perca fluviatilis* (L.)	Velenjsko Lake	0.06	24.9	-	-	-	[41]

References [35,38]—results expressed in mg/kg dry weight.

**Table 3 ijerph-16-03780-t003:** Fatty acids composition (% of total fatty acids) in muscles of different fish species.

Fatty Acids	Bream *Abramis brama* L.		Roach *Rutilus rutilus* L.		Pike *Esox lucius* L.		Perch *Perca fluviatilis* L.	
	Mean	SD	Mean	SD	Mean	SD	Mean	SD
n	5		9		6		8	
C12:0	0.43 ^a^	0.72	0.10 ^a^	0.02	0.13 ^a^	0.01	0.11 ^a^	0.04
C14:0	1.32 ^a^	0.46	1.19 ^a^	0.22	1.30 ^a^	0.37	1.57 ^a^	0.27
C15:0	0.73 ^a^	0.33	0.45 ^b^	0.07	0.43 ^b^	0.04	0.42 ^b^	0.08
C16:0	22.31 ^a^	1.43	22.74 ^a^	1.53	23.36 ^a^	1.98	22.55 ^a^	4.83
C17:0	1.03 ^a^	0.68	0.59 ^b^	0.13	0.48 ^b^	0.06	0.62 ^b^	0.06
C18:0	6.94 ^a^	1.26	6.55 ^a^	0.78	6.57 ^a^	0.38	6.81 ^a^	1.51
C20:0	0.18 ^a^	0.03	0.11 ^b^	0.03	0.08 ^b^	0.02	0.16 ^a^	0.05
C14:1	0.05 ^a^	0.05	0.04 ^ab^	0.03	0.00 ^b^	0.00	0.03 ^ab^	0.04
C16:1	6.03 ^a^	3.43	4.81 ^ab^	1.73	1.94 ^c^	0.36	3.52 ^bc^	0.90
C17:1	0.75 ^a^	0.20	0.46 ^b^	0.19	0.37 ^b^	0.09	0.49 ^b^	0.06
C18:1	16.32 ^a^	6.68	12.93 ^ab^	3.47	9.85 ^b^	1.40	12.11 ^ab^	2.78
C20:1 (n-7)	0.20 ^a^	0.05	0.22 ^a^	0.05	0.12 ^b^	0.01	0.11 ^b^	0.03
C20:1 (n-9)	0.46 ^a^	0.24	0.52 ^a^	0.18	0.39 ^a^	0.19	0.32 ^a^	0.12
C20:1 (n-11)	0.31 ^a^	0.10	0.43 ^a^	0.19	0.05 ^b^	0.08	0.00 ^b^	0.00
C18:2(n-6)	3.36 ^a^	0.54	2.28 ^a^	0.76	2.95 ^a^	0.24	3.13 ^a^	1.83
C18:3γ-lin (n-6)	0.32 ^a^	0.08	0.20 ^b^	0.06	0.21 ^b^	0.02	0.30 ^a^	0.06
C20:2(n-6)	0.70 ^a^	0.19	0.67 ^a^	0.22	0.56 ^a^	0.09	0.24 ^b^	0.10
C20:3(n-6)	0.45 ^a^	0.08	0.38 ^a^	0.10	0.14 ^b^	0.06	0.21 ^b^	0.12
C20:4(n-6)	7.63 ^ab^	2.03	8.51 ^a^	1.93	6.44 ^b^	1.85	8.71 ^a^	0.80
C22:5(n-6)	1.10 ^b^	0.50	2.94 ^a^	1.43	2.03 ^ab^	0.49	1.57 ^b^	0.47
C18:3(n-3)	1.40 ^a^	0.49	1.24 ^a^	0.67	1.68 ^a^	0.29	2.03 ^a^	1.76
C18:4 (n-3)	0.19 ^b^	0.12	0.31 ^b^	0.21	0.75 ^a^	0.11	0.68 ^a^	0.25
C20:3(n-3)	0.48 ^a^	0.29	0.42 ^a^	0.16	0.22 ^b^	0.06	0.23 ^b^	0.09
C20:4(n-3)	0.54 ^a^	0.10	0.81 ^a^	0.44	0.59 ^a^	0.07	0.74 ^a^	0.29
C20:5(n-3) EPA	8.43 ^a^	0.49	7.12 ^a^	2.30	7.36 ^a^	1.25	9.26 ^a^	4.55
C22:5(n-3)	2.32 ^b^	0.47	2.72 ^ab^	0.29	3.00 ^ab^	0.32	3.20 ^a^	1.07
C22:6(n-3) DHA	16.03 ^b^	4.63	21.30 ^b^	3.41	29.02 ^a^	3.61	20.88 ^b^	5.70
Σ SFA	32.94 ^a^	3.28	31.73 ^a^	2.07	32.34 ^a^	2.47	32.24 ^a^	6.33
Σ MUFA	24.12 ^a^	10.30	19.40 ^ab^	5.45	12.73 ^b^	1.76	16.58 ^b^	3.60
Σ n-6 PUFA	13.55 ^a^	2.34	14.98 ^a^	2.63	12.32 ^a^	2.32	14.15 ^a^	2.16
Σ n-3 PUFA	29.39 ^c^	5.90	33.91 ^b^	3.25	42.61 ^a^	3.92	37.02 ^b^	1.82
Σ PUFA	42.94 ^c^	8.19	48.89 ^b^	4.24	54.93 ^a^	3.69	51.17 ^ab^	3.22
n-3/n-6	2.16 ^b^	0.12	2.32 ^b^	0.42	3.56 ^a^	0.67	2.66 ^b^	0.35

SD—standard deviation; ^a, b, c^—significant differences (*p* ≤ 0.05). The same letter (in rows) indicates the absence of significant differences (*p* > 0.05), Ʃ SFA (saturated fatty acid), Ʃ MUFA (monounsaturated fatty acid), Ʃ n-6 PUFA (polyunsaturated fatty acid), Ʃ n-3 PUFA (polyunsaturated fatty acid), EPA-eicosapentaenoic acid (C20:5), DHA-docosahexaenoic acid (C22:6).

**Table 4 ijerph-16-03780-t004:** Estimated daily intake EDI (μg/kg body weight/day).

	Cu	Zn	Mn	Fe	Hg
*Abramis brama* L. (n = 5)	0.117	2.330	0.068	0.729	0.036
*Rutilus rutilus* L. (n = 9)	0.116	2.519	0.059	0.560	0.078
*Esox lucius* L. (n = 6)	0.086	5.582	0.050	0.489	0.165
*Perca fluviatilis* L. (n = 8)	0.076	2.424	0.027	0.574	0.193

**Table 5 ijerph-16-03780-t005:** The hazard quotient calculated for metals content in the muscle tissue of fish.

	Cu	Zn	Mn	Fe	Hg		
RfD (mg/kg/day)	4.00 × 10^−2^	3.00 × 10^−1^	1.4 × 10^−1^	7.00 × 10^−1^	3.00 × 10^−4^		
	THQ					TTHQ	HI
*Abramis brama* L. (n = 5)	0.0029	0.0078	0.0005	0.0010	0.1200	0.132	**1.630**
*Rutilus rutilus* L. (n = 9)	0.0029	0.0084	0.0004	0.0008	0.2596	0.272	
*Esox lucius* L. (n = 6)	0.0022	0.0186	0.0004	0.0007	0.5512	0.573	
*Perca fluviatilis* L. (n = 8)	0.0019	0.0081	0.0002	0.0008	0.6418	0.653	
TDHQ	0.010	0.043	0.0015	0.003	1.573		

n—number of fish; THQ—Target Hazard Quotient; RfD—Oral reference dose (mg/kg/day) [33]; TDHQ—individual toxicant; TTHQ—individual foodstuff; HI—Hazardous Index.

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
