# Peer review of "Health Risk Assessment of Heavy Metals and Lipid Quality Indexes in Freshwater Fish from Lakes of Warmia and Mazury Region, Poland"

_ijerph, 2019, doi:10.3390/ijerph16193780_

Round 1

Reviewer 1 Report

The Latin names of the fish should be changed to italic.
Line 79, 82 I propose that "identification of differences" be added
Line 91 - there should be no reference to the results, it will be in the next chapter.
line 93 - o - should be in the top index
Line 103 - 3 - should be in the lower index
In some places it seems that there are double spaces - line 98, 144, 195, 278, 284, 308, 344, 371
Line 131 - The lipids were extracted according to the Folch's procedure - it would be good to add literature

Line 141-142 -2.5. " were calculated using the following pattern by Ulbricht and Southgate [26], Garaffo et al. [27] and Telahigue et al. [28]: " should be a line below and not italic
Line 144 2.5.1." Index of Atherogenicity (AI): AI = [C12:0+(4 x C14:0)+C16:0]/( n-3PUFA+
145 n-6PUFA+MUFA)" it should be:
"Index of Atherogenicity (AI):
AI = [C12:0+(4 x C14:0)+C16:0]/( n-3PUFA+145 n-6PUFA+MUFA)"
similarly there should be a line below all formulas - line 146, 149, 152, 153

line 169 - improve the formula - minus in the top index
line 176 - double parenthesis - eliminate

Line 221, 223 - draw the right side in the drawings
Line 261 - it would be good to try to explain the reasons for obtaining other results
Line 330 - Adjust
Line 340 - I'm not sure about the grammatical correctness of this sentence - in my opinion it should be "according to this" and before Values and Risk it shouldn't be the same.

Line 377-379 - I think these last two sentences are here by mistake

Line 397 - There was no clear relationship between the tested heavy metals, fatty acids or selected quality indicators and the fish species or their occurrence in organs and biometric data. - this sentence should also be used in the discussion of the results

Author Response

Response to Reviewer 1 Comments

The Latin names of the fish should be changed to italic.

Response 1: The Latin names of the fish were changed as suggested Reviewer #1.

Line 79, 82 I propose that "identification of differences" be added

Response 2: The sentence was added as suggested Reviewer #1.

Line 91 - there should be no reference to the results, it will be in the next chapter.

Response 3: The sentence was changed as suggested Reviewer #1.

line 93 - o - should be in the top index

Response 4: The index was changed as suggested Reviewer #1.

Line 103 - 3 - should be in the lower index

Response 5: The index was changed as suggested Reviewer #1.

In some places it seems that there are double spaces - line 98, 144, 195, 278, 284, 308, 344, 371

Response 6: Double spaces were changed as suggested Reviewer #1.

Line 131 - The lipids were extracted according to the Folch's procedure - it would be good to add literature

Response 7: The reference was added as suggested Reviewer #1.

Line 141-142 -2.5. " were calculated using the following pattern by Ulbricht and Southgate [26], Garaffo et al. [27] and Telahigue et al. [28]: " should be a line below and not italic

Response 8: The sentence was changed as suggested Reviewer #1.

Line 144 2.5.1." Index of Atherogenicity (AI): AI = [C12:0+(4 x C14:0)+C16:0]/( n-3PUFA+

145 n-6PUFA+MUFA)" it should be:

"Index of Atherogenicity (AI):

AI = [C12:0+(4 x C14:0)+C16:0]/( n-3PUFA+145 n-6PUFA+MUFA)"

similarly there should be a line below all formulas - line 146, 149, 152, 153

Response 9: The sentence was changed as suggested Reviewer #1.

line 169 - improve the formula - minus in the top index

Response 10: The index was changed as suggested Reviewer #1.

line 176 - double parenthesis – eliminate

Response 11: The double parenthesis was changed as suggested Reviewer #1.

Line 221, 223 - draw the right side in the drawings

Response 12:  The right side in the drawings was added as suggested Reviewer #1.

Line 261 - it would be good to try to explain the reasons for obtaining other results

Response 13: The reasons for obtaining other results have been explained as suggested Reviewer #1.

Line 330 – Adjust

Response 14: Line has been justified as suggested Reviewer #1.

Line 340 - I'm not sure about the grammatical correctness of this sentence - in my opinion it should be "according to this" and before Values and Risk it shouldn't be the same.

Response 15: The sentence was changed as suggested Reviewer #1.

Line 377-379 - I think these last two sentences are here by mistake

Response 16: The two sentences removed as suggested Reviewer #1.

Line 397 - There was no clear relationship between the tested heavy metals, fatty acids or selected quality indicators and the fish species or their occurrence in organs and biometric data. - this sentence should also be used in the discussion of the results

Response 17: The sentence added to text in section “the discussion of the results” as suggested Reviewer #1.

Reviewer 2 Report

The paper has potential of publishing with the Journal. The design and methods are appropriate and the material presented (fish metal contamination) is in the scope. However, currently, the paper requires substantial language revision and minor clarifications (please see the attached PDF for details). Please, check carefully, there is probably more.

Major comments

1) Methods: Please, provide the area map for the sampling region; What precautions were taken to avoid sample contamination?

2) Results & Discussion: Provide relevant element correlations as diagrams rather than bulky tables. What are the reasons for the observations? Figure 3 is missing.

3) Ethics: I am not sure if an ethical approval is required.

Author Response

Response to Reviewer 2 Comments

The paper has potential of publishing with the Journal. The design and methods are appropriate and the material presented (fish metal contamination) is in the scope. However, currently, the paper requires substantial language revision and minor clarifications (please see the attached PDF for details). Please, check carefully, there is probably more.

Response 1: The manuscript was corrected by a professional language editor as suggested Reviewer #2.

Major comments

1) Methods: Please, provide the area map for the sampling region; What precautions were taken to avoid sample contamination?

Response 2: The area map was added as suggested Reviewer #2.

In the "Methods and material" section, information has been introduced on what precautions have been taken to avoid contamination of the sample: To avoid sample contamination, all organs were collected with a plastic knife and stored in sealed plastic bags.

2) Results & Discussion: Provide relevant element correlations as diagrams rather than bulky tables. What are the reasons for the observations? Figure 3 is missing.

Response 3: The diagram was provided as suggested Reviewer #2.

In the “Results and discussion” section added “The obtained results could have been influenced by the variety of conditions of the aquatic environment in which the fish lived, related to the water reaction and the availability of metals.”

3) Ethics: I am not sure if an ethical approval is required.

Response 4: In the "Methods and material" section information has been introduced

Ethical permit: Fish were bought at the Fish Farm, they were already dead. According to European and Polish Law, the research done on the commercially caught fishes tissue is free to obtain permission on Local Ethical Commision.

Response 5: The data in the Table 1 are given according to the accuracy with which they were marked, as well as the measurement of body length and weight.

Response 6: All comments made by the Reviewer#2 in the comments and in the text have been taken into account and marked.

Round 2

Reviewer 2 Report

I thank the authors for adequately improving their paper in line with my comments